# CAN FOUNDATION MODELS SMELL LIKE HUMANS?

**Farzaneh Taleb, Miguel Vasco, Nona Rajabi, Mårten Björkman, Danica Kragic**
KTH Royal Institute of Technology
Stockholm, Sweden
`{farzantn,miguelsv,nonar,celle,dani}@kth.se`

## ABSTRACT

The human brain encodes stimuli from the environment into representations that form a sensory perception of the world. Despite recent advances in understanding visual and auditory perception, olfaction remains an under-explored topic in the machine learning community due to the lack of large-scale datasets annotated with labels related to human olfactory perception. Simultaneously, foundation models have recently demonstrated impressive performance in several tasks by leveraging large-scale datasets without a supervision signal. In this work, we ask the question of whether foundation models of chemical structures encode representations that are aligned with the human olfactory perception, i.e., *can foundation models smell like humans*? We demonstrate that representations encoded from foundation models pre-trained on general chemical structures are highly aligned with human olfactory perception.

## 1 INTRODUCTION

The human brain receives stimuli from the environment and encodes them into a high-dimensional representation space that forms a sensory perception of the world (Damasio, 1989; Meyer & Damasio, 2009). Sensory modalities such as visual and auditory perception have been extensively explored by both neuroscience and machine learning communities (Sucholutsky et al., 2023; Brohan et al., 2023; Du et al., 2022; Ganis et al., 2004; Friederici, 2012). Findings of such studies highlight a significant level of correlation between human response (from neuron to behavior) and activations of deep neural networks when provided with the same stimuli (Oota et al., 2023; Tang et al., 2023; Dong & Toneva, 2023).

Despite recent advances in understanding visual and auditory perception, human olfaction remains an under-explored topic, regardless of its significant importance in many aspects Iravani et al. (2022); Olsson et al. (2014); Gerkin et al. (2021); Parma et al. (2020). There is no single organizing principle that determines the dimensions of odor space, making the characterization of odor perception and its relation to chemical compounds an open and complex problem (Pannunzi & Nowotny, 2019). Additionally, due to the lack of a universally accepted method to describe odorants either quantitatively or qualitatively, researchers can only rely on the similarity measures (Keller & Vosshall, 2016; Snitz et al., 2013).

There are very few studies that have explored the mapping of chemical structures to olfactory perception. Also, processing chemical olfactory stimuli using deep neural networks has not been extensively investigated. Nevertheless, training the existing models usually requires an extensive (and non-trivial) effort of labeling data by experts. Foundation models (Bommasani et al., 2021) are a recent breakthrough, surpassing the need for extensive labeling by utilizing implicit supervision without the necessity for direct labels. These models have demonstrated impressive performance in various tasks such as image (Dosovitskiy et al., 2020), video (Tong et al., 2022), and natural language processing (Brown et al., 2020).

In this paper, we ask the question of whether representations of odorant chemical structures extracted from foundation models align with human olfactory perception or, in other words, *can foundation models smell like humans*? Specifically, we utilize the representations encoded by MoLFormer (Ross et al., 2022), a state-of-the-art foundation model for chemical structures. To the best of our knowledge, we provide the first empirical study on evaluating the alignment between odorant chemical

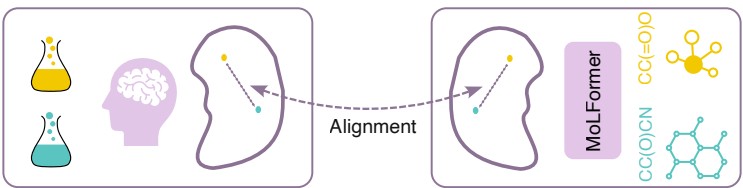

Figure 1: Evaluating the alignment between representations of odorant chemical structures and representations of human olfactory perception. Left: Human participants are stimulated with two odorant substances and asked to rate the perceptual similarity between them. Right: We encode representations of the same odorants using MoLFormer and compute the cosine similarity between pairs of representations. We compute the Pearson correlation between the two similarity measures to quantify their alignment.

representations encoded by foundation models and human perception of odorants. In particular, we show that:

- Representations encoded from foundation models pre-trained on general chemical structures are highly correlated with human olfactory perception *without being explicitly trained for this purpose*;

- Linearly transforming representations extracted from pre-trained foundation models does not significantly improve their alignment with human olfactory perception;

- The level of alignment between pre-trained models on chemical odorants and human olfactory perception increases for representations extracted from deeper layers of the foundation model.

## 2  RELATED WORK

Learning predictive models of olfaction from molecular structures has been addressed mostly by the neuroscience community. Ravia et al. (2020),Keller et al. (2017), and Snitz et al. (2013) used standard chemoinformatic representations of molecules to model olfactory perception. Specifically, Snitz et al. (2013) proposed a computational framework and algorithm based on structural features of molecules to predict perceptual similarities between odorant pairs. This algorithm leverages feature engineering to identify the most relevant subset of features among 1433 chemical descriptors to predict pair-wise odorant perceptual similarities. Later, Ravia et al. (2020) extended this model to also include the intensity of odorant molecules. They employed 21 physicochemical descriptors discovered in previous works and proposed a weighting approach for multicomponent odorants (MC-odorants) based on their perceived intensity. They reported a higher correlation when employing the weighting approach compared to using the same model without it. However, the representation and generalization capabilities of these models are quite limited.

Recently, Lee et al. (2022) proposed a novel representation learning model of odorants, based on a message-passing graph neural network (Gilmer et al., 2017), which they refer to as Principal Odor Map (POM). To train this model, they curated and merged data from Leffingwell (Leffingwell & Associates, 2001) and GoodScent (GoodScent) to create a dataset of about 5000 molecules with 138 expert-labeled odor descriptors. This model outperforms the baselines in multiple odor prediction tasks and shows a relatively high alignment with human ratings in describing odorants. Nevertheless, training this model requires labeled data, relying on subjective evaluations of numerous odorants by experts. Besides being time-consuming and laborious, this process can introduce subjective biases into the model, a concern magnified by our incomplete understanding of the foundational factors of odorants. Large-scale pre-trained models, often known as foundation models (Bommasani et al., 2021), have been recently explored to perform diverse tasks by leveraging large amounts of unlabeled data. The MoLFormer (Ross et al., 2022) model has been proposed in the context of chemical prediction tasks, able to extract rich representations from chemical structures. MoLFormer consists of a transformer-based architecture, with linear attention and relative positional encodings. This model is trained using multiple datasets (e.g., the PubChem (Kim et al., 2019) and ZINC (Irwin & Shoichet, 2005) datasets) on a masked token prediction loss.

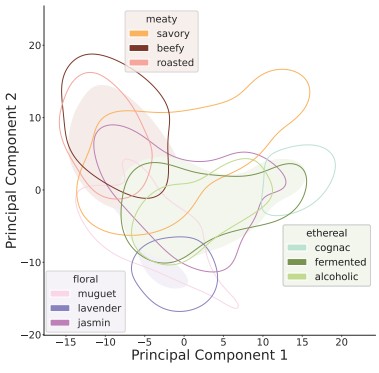 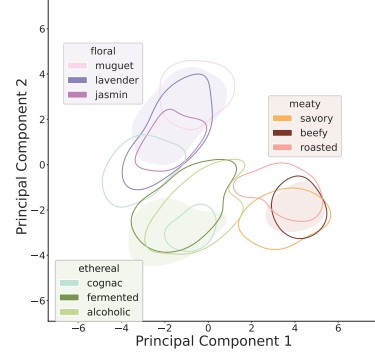

(a) MoLFormer Ross et al. (2022)  (b) POM Lee et al. (2022)

Figure 2: Visualization of odorant representations encoded by different models on the Leffingwell-Goodscent dataset. We plot the first and second principal components (PCs) of the representation spaces. For POM, we replicate the results and use the figure layout of Lee et al. (2022). We highlight the similarity between the representations encoded by both POM and MoLFormer: areas dense with molecules that have broad category labels (floral, meaty, or ethereal) are shaded and areas dense with narrow category labels are outlined. MolFormer is able to capture the perceptual relationship between different odorants in its representation space, despite not being explicitly trained for such purpose.

In this work, we explore MoLFormer for olfactory perception and address the question of whether the representations encoded by this foundation model are aligned with human olfactory perception.

## 3 METHOD

As highlighted in Figure 1, we focus our work on evaluating the alignment between representations of odorant chemical structures, encoded by foundation models, and representations of human olfactory perception.

**Odorants Representations**: Odorants can be described as a single molecule or as a mixture of molecules with varying intensities, which we denote as *multicomponent odorants* (MC-odorants). To represent both types of odorants, we use the simplified molecular-input line-entry system (SMILES), a string-based representation that encodes relevant chemical information such as the type of atoms, their bonds and the substructures present in the molecule.

We employ MoLFormer (Ross et al., 2022) to encode SMILES strings associated with a single molecule and extract a 768-dimensional vector from the last layer of the model. For MC-odorants, we concatenate the input SMILES strings and directly pass the concatenated string to the model. To measure the similarity between pairs of molecules we separately encode each SMILES string and compute the cosine similarity between the extracted representations.

**Perceptual Representations**: Perceptual representations of odorants are provided by human participants when exposed to odorant stimuli. Perceptual olfactory data is often collected in two ways: i) participants are asked to provide ratings in regards to a predefined number of descriptors (Leffingwell & Associates, 2001; GoodScent; Keller & Vosshall, 2016); ii) participants are asked to evaluate the perceived similarity between pairs of odorants (Snitz et al., 2013; Ravia et al., 2020).

**Alignment Function**: To measure the alignment between human olfactory perception and embeddings extracted from MoLFormer we compute the Pearson correlation between *similarity ratings provided by human participants* and *cosine similarity between pairs of odorants embeddings*.

**Evaluation**: Our goal is to evaluate whether pre-trained foundation models of chemical structures encode representations that are aligned with human olfactory perception, despite *not being trained explicitly for that purpose*. In particular, we ask the following questions:

1. Are representations of odorants encoded from foundation models *aligned* with human olfactory perception? (Section 5.1)
2. Can we improve the alignment between representations of foundation models and human perception through linear transformations? (Section 5.2)

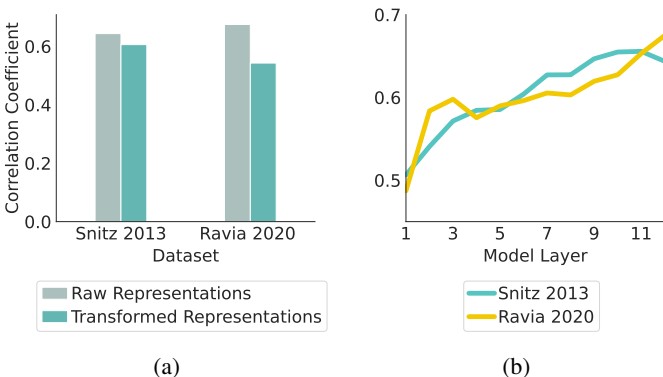

(a)                                                        (b)

Figure 3: Results on the correlation between odorant representations encoded by the MoLFormer model and human olfactory perception from the `Snitz` and `Ravia` datasets: a) Correlation coefficients for original and linearly transformed representations; b) Correlation coefficients considering odorant representations extracted from different depths of the MoLFormer model.

3. How does the degree of alignment change as a function of the representation depth in the foundation model? (Section 5.3)

## 4  DATASETS

We consider multiple datasets containing odorants associated with labels or perceptual similarity ratings:

`Leffingwell-Goodscent` (Leffingwell & Associates, 2001; GoodScent): We employ a curated and merged version of the Leffingwell-Goodscent datasets, provided by Aryan Amit Barsainyan (2023), following the procedure introduced by  Lee et al. (2022). This dataset contains approximately 5,000 molecules with 138 expert-labeled descriptors, where each odorant may be linked to multiple descriptors.

`Snitz` (Snitz et al., 2013): This dataset includes similarity ratings from 139 participants and 359 unique pairs of odorants. Participants rated the perceptual similarity between pairs of odorants. These ratings were then averaged across all subjects. For more details, please refer to the original work.

`Ravia` (Ravia et al., 2020): This dataset contains similarity ratings for 94 participants and 195 unique pairs of MC-odorants and mono-molecules. The similarity values were averaged across all the subjects. In this work, we excluded the factor of intensity of the odorant and aggregated similarity ratings based on the unique pairs of molecules. For more details, please refer to the original work.

## 5  RESULTS

We begin in Section 5.1 by quantitatively evaluating the alignment between representations encoded by the MoLFormer model and human olfactory perception. Additionally, we demonstrate qualitatively that MoLFormer organizes odorant data similar to that of a supervised learning model trained explicitly on odorant classes. In Section 5.2, we present quantitative evidence indicating that traditional probing methods do not significantly alter the alignment level. Lastly, in Section 5.3, we show that the alignment with perceptual representations increases as we extract representations from the network's deeper layers.

### 5.1  REPRESENTATIONS OF ODORANTS EXTRACTED FROM FOUNDATION MODELS ARE HIGHLY ALIGNED WITH HUMAN PERCEPTION OF ODORANTS.

We start by evaluating the alignment between the odorant similarities encoded by MoLFormer and similarity ratings from the `Snitz` and `Ravia` datasets. We compute the Pearson correlation between them and present the results in Figure 3a.

The results show that the MoLFormer is able to extract representations that encode information related to the human olfactory perception, despite not having access to that perception during model training. We highlight a significant high correlation between perceptual and odorant representation for the `Snitz` ($r = 0.64, p < 0.0001$) and `Ravia` datasets ($r = 0.66, p < 0.0001$).

We conduct an additional experiment to understand the degree of perceptual detail captured in the odorant representation space of MoLFormer. We compare odorant representations encoded by MoLFormer with the representations encoded by POM (Lee et al., 2022), a supervised model for odorant perception. Figure 2 depicts the first two principal components of the representations. We observe that MoLFormer is able to capture the perceptual relationship between different odorants in its representation space, despite not using any perceptual labels as a supervision signal (unlike POM).

### 5.2 Linear probing does not improve the alignment between odorant representations of foundation models and human olfactory perception.

Peterson et al. (2018) highlighted that transforming deep representations of visual stimuli using a linear transformation improves the alignment of model representations with human perceptual data, a method we refer to as *linear probing* Alain & Bengio (2016). We analyze the impact of linear transformation on odorant representations by training a linear classifier with four fully-connected layers on MoLFormer's 768-dimensional embeddings from the `Leffingwell-Goodscent` dataset to project them into a 138-dimensional space, corresponding to the dimensionality of the odorant descriptors of the `Leffingwell-Goodscent`. After training the classifier to identify odorant descriptors, we proceed to extract representations for the `Snitz` and `Ravia` datasets from MoLFormer, which are then processed by the trained classifier. Finally, we extract representations from the penultimate layer and follow the procedure detail in Section 3 to compute similarities. We present the results in Figure 3.

The results show that introducing a linear transformation does not affect the correlation coefficient for `Snitz` ($r = 0.64, p < 0.001$) and decreases it for `Ravia` ($r = 0.64, p < 0.001$) dataset. In future work, we plan to investigate whether learning the weights of linear transformations using similarity ratings (in addition to label data) can further improve the alignment between representations.

### 5.3 Olfactory Alignment improves with depth in foundation models.

Scaling model depth often improves task performance in domains such as computer vision (Kolesnikov et al., 2020), but it does not necessarily improve alignment with human perceptual representations (Muttenthaler et al., 2022). We evaluate the effect of model depth in the alignment of odorant representations with human olfactory perception. We extract representations from increasingly deeper hidden layers in the model and follow the procedure detailed in Section 3. The results are presented in Figure 3b, highlighting an increase in the correlation coefficient with increased model depth.

## 6 Conclusion

In this work, we focused on an emergent property of foundation models of general chemical structures: the ability to encode representations that are aligned with human olfactory perception. We demonstrated that odorant representations encoded by the MoLFormer model are highly aligned with human olfactory perception. Furthermore, we have shown the effect of linear transformations and network depth on the level of this alignment. We plan on exploring further emergent properties of foundation models of chemical data and human olfactory data, in particular the alignment of odorant representations with recorded data from the brain, such as fMRI.

## Acknowledgement

This work has been supported by the Swedish Research Council, Knut and Alice Wallenberg Foundation, and ERC-2023-Syg 101118977 D2Smell.

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
