# OpenReview forum: "Can Foundation Models Smell Like Humans?"
_ICLR.cc/2024/Workshop/Re-Align — ICLR 2024 Workshop Re-Align Poster_

### Official Review · Reviewer_h8hr · 2024-02-19

**Rating:** 2
**Fit:** 3
**Confidence:** 2

**Workshop Review:**

**Summary:**

The authors examine the alignment between humans and machines in the field of olfactory perception. A topic that has been rarely researched so far and thus offers an interesting and new insight into human-machine alignment. For this, the authors utilized a state-of-the-art foundation model for chemical structures (MoLFormer) and compared its representations with various well-known datasets in the field. Their results show that the MoLFormer encodes information related to human olfactory perception, and thereby the perceptual relationship between odorants. Further, they showed that linear transformations of these representations do not result in a better alignment, which is the case in other perceptual modalities like vision. Lastly, they can show that olfactory alignment improves with model depth of the foundation model.

---

**Clarity:** low \
**Correctness:** good \
**Novelty:** good \
**Interest to the community:** good

---
**Strong points:**

- **Novel topic** (study focuses on the alignment of foundation models with human olfactory perception)
- **Good experimental setup** (Used well-known datasets in the field, and an adequate state-of-the-art foundation model)
- **Adequate Method** (the study uses adequate measures to compute the alignment between human olfactory perception and representations in the foundation model (MoLFormer)
- **Of interest for the community:** (The study is at the heart of human and machine alignment and thereby an interesting contribution to the community)


**Weak points:**

- **Lacking clarity:** While the paper is well written, it is hard to understand. The connection between olfactory perception and chemical compounds (does structure equal odor perception) is unclear. The sentence describing these studies misses citations (see: *“There are very few studies that have explored the mapping of chemical structures to olfactory perception.”*) As far as I understand it this is crucial as the MoLFormer represents the chemical structures and seems to capture the olfactory perception. The authors do not discuss, interpret, and clarify the impact of their findings.

- **No baselines or comparison:** While other research like the POM (Lee et al., 2022) used baselines to understand better how well olfactory perception is captured by the models here no baseline is tested. This makes it hard to understand and interpret the resulting correlation value and thereby the alignment of the foundation model with human olfactory perception.

- **Clarification of results:** While the results a clearly stated their motivation and benefits are less clear. Especially the sections 5.2 and 5.3 feel motivated by research in other domains but seem to be not motivated by olfactory perception e.g. why would we expect to find that olfactory alignment improves with model depth? Does this indicate that olfactory perception is hierarchical?
The strongest point of this paper is that it finds that foundation models representing chemical structure show alignment with olfactory perception. But is it possible in this paradigm and with these datasets to test edge cases like Sell’s triplets where different structures result in the same olfactory perception?  (Lee et al., 2022).

---
**Questions:**

Question 1:  Do you think/Do your results imply that chemical structure determines olfactory perception? How could this be disentangled in your paradigm? \
Question 2: What kind of alignment between MoLFormer and olfactory perception would you expect with Sell’s triplets (Lee et al., 2022). \
Question 3: Alignment is only tested with the Snitz and Ravia dataset, as described in your method section they contain perceived similarity between pairs. What kind of alignment would you expect with the other methods with predefined descriptors? You used this dataset to test linear transformation but could it be also tested on its own? \
Question 4: Why do you think model depth improves olfactory perception? \
Question 5: Why does the linear transformation for the Ravia dataset decrease the alignment? \
Question 6: What would be suitable baselines to test and compare the alignment against? \

**Reason For Not Giving Higher Score:**

lack in clarity

**Reason For Not Giving Lower Score:**

relevant and novel topic that might be interesting for the community and the work might benefit from feedback during the workshop.

**Reviewer Domain:**

cognitive science

---

### Official Review · Reviewer_nnjm · 2024-02-21
**Though preliminary, results are clear and hold promise towards answering an important and practical question.**

**Rating:** 2
**Fit:** 3
**Confidence:** 2

**Workshop Review:**

In this work the author’s ask whether Foundational Models trained without explicit human feedback learn similar olfactory representations to humans. This question is deep and important for multiple reasons but especially because the principles by which humans perceive odor similarity or dissimilarity are presently unknown. Knowing whether foundation models perceive odors similarly to humans would allow the field to search for organizing principles in silico using foundation models and then test the predictions in humans.

To test this question, the author’s compared the responses of MolFormer (a foundation model using the masked token self-supervised loss) first to a model trained to a model trained via supervised learning, and finally to two human odor similarity datasets. The author’s found, surprisingly, that MolFormer seems to represent odors similarly to the model trained via supervised learning, despite MolFormer being trained on a general set of chemicals and on an unrelated loss. The authors further found that odor similarity within MolFormer algins with human odor similarity measurements.

The claims made in this study are well supported by the data, and the motivations of the analyses are clear. This work has many strengths and great promise to be better in the future, but at present some of the analyses are superficial and should be expanded on e.g the author’s found that alignment with human metrics increases with model layer depth a sensitivity analysis would be useful here in helping the reader understand what chemical features each layer is sensitive to and how those features relate to handcrafted features previously used for predicting human odor preferences.

**Reason For Not Giving Higher Score:**

The results, while clear and rigorous, are superficial and could provide much more insight with some more work/exploration in the future.

**Reason For Not Giving Lower Score:**

The work is well motivated and relevant to the workshop topic.

**Reviewer Domain:**

neuroscience

---

### Official Review · Reviewer_ba6Z · 2024-02-24
**An Interesting Observation, In Need of Baselines**

**Rating:** 1
**Fit:** 3
**Confidence:** 2

**Workshop Review:**

This paper investigates the alignment between a network trained using an MAE style loss on a dataset of chemical structures produces representations that are aligned with human perceptions of odor similarity.

Strengths
- Though this is not my area of expertise I agree with the authors that olfaction is understudied in the ML space, and this work seems like an interesting extension of observations from other sensory domains.

Weaknesses
- The main weakness in this work is a lack of reasonable baselines with which to compare the relative alignment of the human judgements and the self supervised transformer. Some options the authors could consider:

    -  Comparing to POM in figure 3. I understand that POM was trained using human judgements while the MoLFormer was not, but this would be a relevant comparison for evaluating the extent to which alignment "emerges" from the self supervised pretraining.
   - Another comparison of interest would be the alignment between the inputs to the MoLFormer and human similarity jugdements (i.e. adding model layer 0 to figure 3b).

   - Evaluating the alignment of the SSL model during the course of learning

Overall I cannot recommend acceptance as is, though the work certainly has potential to grow into something that would be of significant interest to the Re-Align community. The contribution of the work is not to provide a new model/framework, but to evaluate the alignment of a self supervised model to human similarity judgements. While these measurements are certainly made I am of the meaning that they are not particularly meaningful without comparisons to meaningful baselines.

**Reason For Not Giving Higher Score:**

Lack of relevant baselines.

**Reason For Not Giving Lower Score:**

N/A

**Reviewer Domain:**

neuroscience

---

### Decision · Program_Chairs · 2024-03-02

Accept (Poster)